# Effect of dietary branched chain amino acids on liver related mortality: Results from a large cohort of North American patients with advanced HCV infection

**Lei Yu** [1]*, **Shirley C. Paski**[2], **Jennifer Dodge**[3], **Kiran Bambha**[1], **Scott W. Biggins**[1], **George N. Ioannou**[1,4]

1 Division of Gastroenterology, University of Washington, Seattle, Washington, United States of America, 2 Division of Gastroenterology, Cedar Sinai School of Medicine, Los Angeles, California, United States of America, 3 Department of Surgery, Division of Transplant Surgery, University of California San Francisco, San Francisco, California, United States of America, 4 Veterans Affairs Puget Sound Health Care System, Seattle, Washington, United States of America

* leiy@medicine.washington.edu

**Data Availability Statement:** The data is public from the National Institutes of Digestive Disease

## Abstract

Branched chain amino acids (BCAA) supplementation may reduce the incidence of liver failure and hepatocellular carcinoma in patients with cirrhosis. We aimed to determine whether long-term dietary intake of BCAA is associated with liver-related mortality in a well-characterized cohort of North American patients with advanced fibrosis or compensated cirrhosis. We performed a retrospective cohort study using extended follow-up data from the Hepatitis C Antiviral Long-term Treatment against Cirrhosis (HALT–C) Trial. The analysis included 656 patients who completed two Food Frequency Questionnaires. The primary exposure was BCAA intake measured in grams (g) per 1000 kilocalories (kcal) of energy intake (range 3.0–34.8 g/1000 kcal). During a median follow-up of 5.0 years, the incidence of liver-related death or transplantation was not significantly different among the four quartiles of BCAA intake before and after adjustment of confounders (AHR 1.02, 95% CI 0.81–1.27, P-value for trend = 0.89). There remains no association when BCAA was modeled as a ratio of BCAA to total protein intake or as absolute BCAA intake. Finally, BCAA intake was not associated with the risk of hepatocellular carcinoma, encephalopathy or clinical hepatic decompensation. We concluded that dietary BCAA intake was not associated with liver-related outcomes in HCV-infected patients with advanced fibrosis or compensated cirrhosis. The precise effect of BCAA in patients with liver disease warrants further study.

## Introduction

Leucine, isoleucine and valine, collectively known as the branched chain amino acids (BCAA), have been the subject of intense investigation in liver disease. Compared to normal controls, patients with a wide range of liver disease etiologies and severity have significantly lower plasma concentrations of the BCAA and higher concentrations of the aromatic amino acids

and Kidney Disease (NIDDK, NIH, United States). https://repository.niddk.nih.gov/user/login.

**Funding:** C-LIFE, Center for Liver Investigation Fostering Discovery. The funders had no role in study design, data collection and analysis, decision to publish, or preparation of the manuscript.

**Competing interests:** The authors have declared that no competing interests exist.

**Abbreviations:** AHR, Adjusted hazard ratio; AST, Aspartate transaminase; ALT, alanine transaminase; BCAA, Branched chain amino acids; CTP, Child-Turcotte-Pugh; FFQ, food frequency questionnaire; HR, hazard ratio; HCC, hepatocellular carcinoma; HCV, hepatitis C virus; INR, international normalized ratio; IQR, interquartile range; IU, international units; SD, standard deviation.

(AAA) tyrosine and phenylalanine [1, 2]. This imbalance of plasma amino acids correlated with histological severity and portal systemic shunting [1, 3], but not with serum biochemical parameters or the presence of hepatic encephalopathy [1]. From a therapeutic perspective, the majority of investigations have focused on the effect of BCAA on hepatic encephalopathy. The rationale was proposed by James et al. who theorized that BCAA would compete with AAA for entry into the brain, thereby limiting the formation of "false" neurotransmitters [4]. While most short-term randomized control trials of intravenous BCAA showed benefits in acute episodes of encephalopathy [5], long-term oral formulations in chronic encephalopathy were associated with mixed results [6–8]. More recently, beneficial effects of BCAA have been expanded to the prevention of liver failure, hepatocellular carcinoma (HCC) and its recurrence following treatment [6, 9–14]. The most plausible mechanisms include the ability of BCAA, in particular leucine, to stimulate liver regeneration via the production of hepatocyte growth factor and their ability to improve insulin resistance–a key driver in hepatic fibrogenesis and carcinogenesis [15–18]. In Japan, BCAA is an approved pharmaconutrient for patients with cirrhosis [19, 20].

Even though BCAA are readily available in the diet, to our knowledge, no study has closely examined dietary BCAA intake in cirrhotic patients and whether the variability in intake affects disease progression. While several of these aforementioned trials attempted to control dietary protein intake [6, 7, 10], none specifically estimated or accounted for dietary BCAA in their analysis. In the current study, we aimed to determine whether long-term intake of BCAA from diet is associated with liver-related mortality in patients with advanced fibrosis or compensated cirrhosis.

## Subjects and methods

### Study design and ethics statement

We performed a retrospective cohort study using extended long-term follow-up data from the Hepatitis C Antiviral Long-term Treatment against Cirrhosis (HALT-C) Trial (ClinicalTrials.gov #NCT00006164). The trial included HCV-infected patients with histologically defined advanced fibrosis (Ishak fibrosis score 3 or 4) or compensated cirrhosis (Ishak fibrosis score 5 or 6). No patients had a history of Child-Turcotte-Pugh (CTP) score ≥7, bleeding related to esophageal varices, ascites, encephalopathy or HCC at the time of entry into the study. Patients were followed for a median duration of 6 years for clinical decompensation, HCC and death. The data generated by the HALT-C study have been used to evaluate multiple risk factors for liver disease progression in HCV-infected patients [21, 22]. Details on the design, patient population and study outcomes have been published elsewhere [23, 24]. All study subjects provided written informed consent and the study protocol were approved by Institutional Review Boards of all participating centers. The current study analyzed deidentified trial data provided by the National Institutes of Health. The University of Washington Institutional Review Board waived the requirement for informed consent.

### Assessment of dietary branched chain amino acids: Leucine, isoleucine and valine

Dietary intake was assessed using the well-validated Block 98.2 Food Frequency Questionnaire (FFQ, Block Data Systems, Berkeley, California), which estimates nutrient intake over the past year based on self-reported frequency and portion of foods [25]. The Block FFQ estimates the amount of specific nutrients primarily based on the United States Department of Agriculture (USDA) Database for Standard Reference, Release 27. These nutrient values were then

combined with population and consumption-weighted intake data from NHANES (2007–2010) 24-hour dietary recalls, in order to determine the amount of nutrients found in the food or beverage in the questionnaire response. The average intake of a nutrient from a specific food item is calculated as the product of reported frequency of intake, reported portion size and the estimated nutrient amount from USDA and NHANES. Out of 1,050 patients who participated in the randomized phase of the study, 808 patients completed FFQ at enrollment, 822 patients completed a follow-up FFQ 661±269 days after enrollment and 672 patients completed both FFQs. Because diet may change over time, the average nutrient intake, calculated using multiple assessments, can capture dietary intake more accurately than a single assessment [26]. We therefore defined BCAA intake as the average intake estimated from the baseline and the follow-up FFQ's calculated as following: (BCAA from baseline FFQ in grams + BCAA from follow-up FFQ in grams)/2.

## Study population

Among 672 patients who completed both FFQ's, we excluded one patient who reported an extremely high caloric intake (more than 2 interquartile ranges from the median). We also excluded 15 patients who underwent transplantation before the second FFQ (N = 2), whose date of completion of the second FFQ was unknown (N = 10), who enrolled greater than 1 year prior to completing the second FFQ (N = 1), and who did not have any follow-up after completing their second FFQ (N = 2), leaving 656 patients in the current analysis.

## Outcome definition

The primary outcome of the study is the development of either liver-related death or liver transplantation from the time of the completion of the follow-up FFQ. Secondary outcomes are the development of 1. HCC, 2. hepatic encephalopathy or 3. a composite endpoint of first clinical decompensation including ascites, spontaneous peritonitis, variceal bleeding or encephalopathy. Liver-related death is defined as death from end stage liver disease progression, liver cancer or Peg-interferon treatment induced injury as judged by a 7-person central review committee using the following likelihood categories: "unlikely" or <25% likelihood, "possible" or 25–49% likelihood, "probable" or 50–75% likelihood, and "highly likely" or >75% likelihood. When the likelihood is in the "probable" or "highly likely" categories, the cause of death would be considered as liver-related [24].

## Statistical analysis

We used Cox proportional-hazards regression to determine whether total BCAA consumption was associated with liver-related death or transplantation. Since total BCAA intake was positively correlated with total energy intake (Pearson's correlation: r = 0.771, p < 0.001), quartiles of total BCAA intake were created after dividing BCAA intake by total energy intake, according to the nutrient density model of total energy adjustment [27]. BCAA intake quartiles were modeled in two ways: 1. As a "dummy" categorical variable, where each higher quartile of intake (2nd, 3rd and 4th) was compared to the lowest (1st) quartile; and 2. As an ordinal variable (1 through 4) to yield a "test for trend" where each higher quartile of BCAA intake was compared to the lower quartile. Patients were censored at their last clinic visit. We performed multivariable Cox regression analyses adjusting for the following variables that may influence disease progression in chronic liver diseases: age, gender, race, body mass index (BMI), diabetes, lifetime alcohol consumption, smoking status, self-reported health status (excellent, very good, good, fair and poor), coffee intake (nondrinker, <1, 1–2 and ≥3 cups/day), duration of HCV infection, treatment with low-dose peginterferon, presence of cirrhosis, total energy

intake and cholesterol intake. Variables were modeled linearly except for gender, race, diabetes, self-reported health status, low-dose peginterferon treatment group, current smoking status, cirrhosis status and coffee intake. The assumption of proportional hazards was tested and met using weighted residual methods.

We performed sensitivity analyses using two additional methods of capturing BCAA intake as an exposure variable. First, because total BCAA intake was positively correlated with total amino acids (protein) intake (Pearson's correlation: $r = 0.859$, $p < 0.001$), and different amino acids are metabolized differently [28], particularly in advanced cirrhosis [29], we hypothesize that the "ratio" of BCAA to total protein intake may have a different effect on the incidence of liver-related death or liver transplantation. This ratio, "BCAA/total protein," was first obtained by dividing the average BCAA intake by the average total protein intake. Quartiles of BCAA/Protein intake were then created by dividing "BCAA/total protein" by total energy intake according to the nutrient density model of total energy adjustment as we did above for our primary analysis. Cox regression models included the same variables as the primary analysis. Second, because the true relationship between dietary BCAA and liver disease outcomes may not be affected by total energy intake, we aimed to assess whether the absolute intake of BCAA affected liver-related death or transplantation. Quartiles of absolute BCAA intake were created using the average intake of BCAA from both FFQ's. In this model, Cox regression models included the same variables as the primary analysis except total energy intake. Sample size was pre-determined by the HALT-C trial. Analyses were performed with SAS version 9.4 (Cary, NC), Stata/IC 14.2 and Stata/SE 11.0 (College Station, TX).

## Results

The quartiles of average BCAA intake among 656 patients with advanced fibrosis or compensated cirrhosis were categorized as follows: < 5.6 g/1000 kcal, 5.6–6.4 g/1000 kcal, 6.4–7.4 g/1000 kcal and > 7.4 g/1000 kcal. The median absolute BCAA intake for the quartiles were 9.6 g/day (IQR 6.2–12.6 g/day), 11.7 g/day (IQR 8.2–14.4 g/day), 12.4 g/day (IQR 9.4–16.5 g/day) and 15.7 g/day (IQR 11.2–21.4 g/day), respectively. The median absolute BCAA intake for the entire study cohort was 11.9 g/day (IQR 8.6–16.2 g/day). For the entire cohort, there was no significant change in median BCAA intake estimated from the follow-up FFQ (6.4 g/1000 kcal, IQR 5.4–7.5 g/1000 kcal) compared to that from baseline (6.4 g/1000 kcal, IQR 5.4–7.3 g/1000 kcal). On an individual level, the median difference between BCAA intake estimated from the follow-up and baseline FFQ was only 0.05 g/1000 kcal (IQR -0.9–1.2 g/1000 kcal).

Higher BCAA intake was associated with higher intake of AAA, total protein and cholesterol, but lower intake of carbohydrates, fats and lifetime alcohol. Dietary intake of fiber, vegetables and fruits were not significantly different among the BCAA intake categories (Table 1). In terms of metabolic parameters, higher BCAA intake was associated with higher prevalence of diabetes (Table 2). Higher BCAA intake was not associated with baseline liver enzyme levels or histological scores of inflammation or fibrosis. The proportion of patients with cirrhosis or varices at baseline did not vary significantly among categories of BCAA intake (Table 2).

During a median follow-up of 5.0 years, the incidence of liver-related death (N = 45) or transplantation (N = 52) was 31.5 per 1000 person-years. There were 32 non-liver related deaths during the study period. The incidence of all-cause mortality (liver and non-liver related death) or liver transplantation was 41.9 per 100 person-years. The incidence of HCC (N = 53), hepatic encephalopathy (N = 35) and first clinical hepatic decompensation (N = 74) were 17.2, 11.4 and 25.1 per 1000 person-years, respectively. Rates of individual decompensation events such as variceal bleeding, ascites and bacterial peritonitis are presented in S1 Table. Overall, the incidence of liver-related death or transplantation was not significantly different

**Table 1. Nutrients intake of 656 patients according to quartiles of average dietary BCAA intake (measured in grams per 1000 kcal of daily energy intake).**

| | All patients | Quartiles of BCAA intake (grams per 1000 kcal of daily energy intake) | | | | P-value for trend |
|---|---|---|---|---|---|---|
| | | 1st (< 5.6) | 2nd (5.6–6.4) | 3rd (6.4–7.4) | 4th (>7.4) | |
| N | 656 | 164 | 164 | 164 | 164 | |
| Average Daily Dietary Intake | | | | | | |
| Total energy (Kcal, median, IQR) | 1827 (1343–2349) | 1929 (1322–2524) | 1924 (1358–2422) | 1820 (1381–2362) | 1660 (1291–2227) | 0.02 |
| Carbohydrate (grams, median, IQR) | 227 (163–297) | 249 (185–320) | 245 (165–298) | 215 (160–273) | 203 (147–272) | <0.001 |
| Total fat (grams, median, IQR) | 74.9 (53.5–104.5) | 81.4 (56.5–109.8) | 78.1 (52.6–105.2) | 76.5 (55.9–107.4) | 66.5 (49.2–89.4) | 0.01 |
| Saturated fat (grams, median, IQR) | 23.2 (16.1–32.2) | 24.6 (16.5–34.2) | 24.7 (16.5–32.0) | 23.4 (16.8–34.0) | 21.4 (15.3–28.5) | 0.05 |
| Unsaturated fat (grams, median, IQR) | 45.5 (32.7–63.1) | 49.4 (33.4–65.2) | 47.3 (32.6–65.6) | 47.8 (35.1–66.3) | 41.3 (31.2–53.5) | 0.04 |
| Trans fat (grams, median, IQR) | 6.1 (4.0–9.6) | 7.2 (4.6–10.7) | 6.4 (4.2–10.2) | 6.3 (4.4–8.8) | 4.8 (3.2–6.9) | <0.001 |
| Protein (grams, median, IQR) | 65.8 (48.0–85.7) | 58.3 (38.7–77.1) | 66.4 (46.8–84.7) | 70.1 (51.5–91.9) | 68.4 (54.3–90.8) | <0.001 |
| Leucine (mg, median, IQR) | 5323 (3823–7222) | 4283 (2768–5575) | 5158 (3656–6385) | 5507 (4179–7295) | 6923 (4949–9366) | <0.001 |
| Isoleucine (mg, median, IQR) | 3094 (2260–4224) | 2535 (1641–3305) | 3039 (2135–3800) | 3245 (2433–4305) | 4071 (2956–5675) | <0.001 |
| Valine (mg, median, IQR) | 3528 (2570–4786) | 2914 (1853–3758) | 3434 (2431–4275) | 3639 (2751–4834) | 4602 (3288–6280) | <0.001 |
| Phenylalanine (mg, median, IQR) | 2958 (2160–4052) | 2462 (1590–3266) | 2917 (2042–3576) | 3044 (2306–4094) | 3741 (2745–5020) | <0.001 |
| Tyrosine (mg, median, IQR) | 2416 (1752–3295) | 1976 (1264–2566) | 2370 (1646–2867) | 2490 (1882–3335) | 3154 (2265–4375) | <0.001 |
| Tryptophan (mg, median, IQR) | 792 (576–1083) | 659 (418–863) | 784 (535–974) | 813 (615–1100) | 977 (740–1396) | <0.001 |
| Dietary fiber (grams, median, IQR) | 15.3 (11.1–20.4) | 15.0 (10.6–20.0) | 15.6 (11.3–20.2) | 15.8 (11.3–21.0) | 14.6 (10.6–21.0) | 0.67 |
| Cholesterol (mg, median, IQR) | 220 (152–311) | 195 (128–273) | 214 (146–300) | 235 (174–331) | 242 (176–339) | <0.001 |
| Daily servings of vegetables | 2.4 (1.5–3.5) | 2.4 (1.4–3.2) | 2.3 (1.5–3.4) | 2.4 (1.6–3.8) | 2.2 (1.6–3.7) | 0.22 |
| Daily servings of fruit and fruit juices | 1.3 (0.8–2.0) | 1.4 (0.8–2.0) | 1.3 (0.8–2.0) | 1.2 (0.8–2.0) | 1.3 (0.8–2.1) | 0.57 |

among the quartiles of BCAA intake before or after adjustment of potential confounders (Fig 1 and Table 3, AHR 1.02, 95% CI 0.81−1.27, P-value for trend = 0.89). Crude and adjusted hazard ratios for other confounding variables are presented in S2 Table. The incidence of HCC (AHR 0.89, 95% CI 0.65−1.21, P-value for trend = 0.45) or hepatic encephalopathy (AHR 0.98, 95% CI 0.68−1.42, P-value for trend = 0.93) were also not significantly different among the groups. Similarly, there was no significant association between BCAA intake categories and first clinical hepatic decompensation (variceal bleeding, ascites, peritonitis or encephalopathy). Details of hazard ratio estimates are presented in S3 Table.

Finally, there was no significant association between BCAA intake and liver-related death or transplantation when BCAA intake was modeled in terms of quartiles of BCAA to total protein ratio (with total energy intake adjustment, AHR 1.05, 95% CI 0.72−1.52, P-value for trend = 0.80, S4 Table) or quartiles of absolute BCAA intake (without total energy intake adjustment, AHR 1.04, 95% CI 0.78−1.41, P-value for trend = 0.77, S5 Table).

## Discussion

Using carefully collected prospective data from the HALT-C trial, we did not find any association between BCAA intake from dietary sources and liver-related death or transplantation (our primary outcome) or development of HCC or hepatic encephalopthy (our secondary outcomes) in hepatitis C infected patients with advanced fibrosis or compensated cirrhosis.

Our negative finding differs from the majority of published reports [6, 9–13, 30]. In the literature, the first strong evidence on the benefit of BCAA came from the Italian double-blind randomized trial in which BCAA supplementation of 14.4 g/day for one year was associated with an improvement in overall outcomes in patients with Child-Pugh class B or C cirrhosis. Compared to two control groups who received lactoalbumin or maltodextrin, patients in the BCAA group had lower rates of death and clinical progression, reduced hospital admission

**Table 2. Baseline characteristics of 656 patients according to quartiles of average dietary BCAA intake (measured in grams per 1000 kcal of daily energy intake).**

| | All patients | Quartiles of BCAA intake (grams per 1000 kcal of daily energy intake) | | | | P-value for trend |
|---|---|---|---|---|---|---|
| | | 1st (<5.6) | 2nd (5.6–6.4) | 3rd (6.4–7.4) | 4th (>7.4) | |
| N | 656 | 164 | 164 | 164 | 164 | |
| Age at randomization (years, median,IQR) | 50 (46–54) | 49 (46–53) | 49 (46–55) | 51 (47–55) | 50 (47–54) | 0.10 |
| Female (%) | 29 | 33 | 24 | 29 | 32 | 0.83 |
| Race (%) | | | | | | 0.63 |
| White | 75 | 72 | 77 | 76 | 76 | |
| Black | 15 | 19 | 14 | 16 | 13 | |
| Hispanic | 7 | 6 | 8 | 6 | 7 | |
| Other | 3 | 3 | 1 | 2 | 4 | |
| BMI (median, IQR) | 29.0 (26.3–32.4) | 29.0 (26.1–31.8) | 28.7 (26.3–32.4) | 28.4 (26.1–32.9) | 29.7 (26.6–33.0) | 0.17 |
| [a]Waist-to-hip ratio (median, IQR) | 0.57 (0.52–0.62) | 0.57 (0.52–0.61) | 0.57 (0.53–0.61) | 0.57 (0.52–0.61) | 0.57 (0.52–0.64) | 0.42 |
| [b]Self-reported health status (%) | | | | | | 0.11 |
| Excellent | 4 | 1 | 4 | 4 | 5 | |
| Very good | 23 | 20 | 25 | 22 | 25 | |
| Good | 45 | 46 | 44 | 50 | 39 | |
| Fair | 24 | 27 | 22 | 18 | 28 | |
| Poor | 5 | 6 | 6 | 6 | 3 | |
| Lifetime alcohol consumption (number of drinks, median, IQR) | 7229 (1096–20778) | 8430 (1397–30240) | 8646 (1056–20560) | 7499 (1443–22216) | 5098 (399–14621) | 0.02 |
| [c]Daily coffee intake (%) | | | | | | 0.58 |
| Non-drinkers | 17 | 19 | 19 | 13 | 20 | |
| <1 cup | 27 | 22 | 31 | 30 | 24 | |
| 1 to 2 cups | 45 | 44 | 39 | 48 | 48 | |
| ≥ 3 cups | 11 | 15 | 11 | 9 | 9 | |
| Current smokers (%) | 27 | 33 | 24 | 26 | 23 | 0.06 |
| [d]Weekly recreational and non-recreational physical activity (metabolic equivalents, median, IQR) | 116 (77–161) | 121 (79–169) | 113 (77–154) | 112 (70–149) | 116 (80–163) | 0.86 |
| Glucose (mg/dL, median, IQR) | 96 (88–113) | 95 (87–108) | 99 (88–116) | 96 (88–114) | 96 (88–113) | 0.35 |
| [e]Insulin (microunit/ml, median, IQR) | 34 (23–57) | 35 (22–59) | 34 (24–55) | 35 (24–56) | 34 (23–56) | 0.83 |
| [f]HOMA2-IR (median, IQR) | 4.2 (2.9–6.8) | 4.2 (2.7–7.0) | 4.2 (2.9–6.7) | 4.3 (3.0–6.7) | 4.1 (2.9–6.4) | 0.80 |
| Diabetes (%) | 15 | 7 | 12 | 16 | 23 | <0.001 |
| Baseline serum triglycerides (mg/dL, median, IQR) | 109 (78–164) | 108 (70–159) | 107 (80–153) | 116 (78–179) | 109 (79–166) | 0.20 |
| Follow-up length (years since follow-up FFQ, median, IQR) | 5.0 (3.9–5.7) | 5.2 (4.0–5.9) | 5.1 (3.6–5.6) | 5.0 (3.7–5.6) | 5.0 (4.0–5.6) | 0.11 |
| Randomized to Peginterferon (%) | 51 | 44 | 53 | 55 | 51 | 0.18 |
| [g]Duration of HCV infection (years, median, IQR) | 29 (24–33) | 30 25–33) | 28 (22–33) | 29 (23–32) | 29 (24–33) | 0.68 |
| HCV Genotype 1 (%) | 93 | 92 | 92 | 95 | 94 | 0.33 |
| HCV Viral load (Log IU/ml, median, IQR) | 6.5 (6.2–6.8) | 6.4 (6.2–6.7) | 6.5 (6.2–6.8) | 6.5 (6.2–6.7) | 6.6 (6.1–6.8) | 0.12 |
| Cirrhosis (%) | 41 | 41 | 42 | 43 | 37 | 0.50 |
| [h]Varices (%) | 26 | 24 | 32 | 26 | 21 | 0.36 |
| ALT (units/L, median, IQR) | 86 (58–134) | 80 (53–133) | 93 (60–133) | 82 (54–128) | 92 (62–138) | 0.44 |
| AST (units/L, median, IQR) | 71 (49–109) | 70 (48–109) | 72 (50–106) | 71 (47–108) | 70 (51–114) | 0.80 |
| Total bilirubin (mg/dL, median, IQR) | 0.7 (0.5–1.0) | 0.7 (0.5–1.0) | 0.7 (0.5–1.0) | 0.7 (0.5–1.0) | 0.7 (0.5–0.9) | 0.24 |
| Albumin (g/dL, median, IQR) | 3.9 (3.6–4.2) | 3.9 (3.6–4.1) | 3.9 (3.7–4.2) | 3.9 (3.6–4.2) | 3.9 (3.6–4.1) | 0.92 |
| Platelets (10³ per mm³, median, IQR) | 162 (116–207) | 165 (112–205) | 160 (114–211) | 157 (120–201) | 161 (126–209) | 0.80 |
| INR (median, IQR) | 1.0 (1.0–1.1) | 1.0 (1.0–1.1) | 1.0 (1.0–1.1) | 1.0 (1.0–1.1) | 1.0 (1.0–1.1) | 0.46 |

*(Continued)*

**Table 2.** (Continued)

| | All patients | Quartiles of BCAA intake (grams per 1000 kcal of daily energy intake) | | | | P-value for trend |
|---|---|---|---|---|---|---|
| | | 1st (<5.6) | 2nd (5.6–6.4) | 3rd (6.4–7.4) | 4th (>7.4) | |
| Ishak inflammation score (median, IQR) | 7 (6–9) | 8 (6–9) | 7 (6–9) | 7 (6–9) | 7 (6–9) | 0.86 |
| Ishak fibrosis score (median, IQR) | 4 (3–5) | 4 (3–5) | 4 (3–5) | 4 (3–5) | 4 (3–5) | 0.64 |

[a] 643 patients had known waist-to-hip ratio status.

[b] 650 patients had known self-reported health status.

[c] 620 patients had known coffee intake estimated from the baseline FFQ.

[d] 500 patients had known weekly self-reported recreational and non-recreational physical activity

[e] 496 patients had baseline insulin measurement.

[f] 481 had baseline HOMA-IR status.

[g] 620 patients had known duration of HCV infection.

[h] 650 patients had known baseline varices status.

rates and length of stay, less cirrhosis-related anorexia and better nutritional parameters [6]. Since then, a second randomized trial from Japan in which Child-Pugh class A cirrhotic patients were also included confirmed a modest benefit of BCAA supplementation of 12 g/day during a two-year study period [10]. Subsequent uncontrolled studies from Japan extended the benefit of BCAA supplementation to the prevention of *de novo* HCC and its recurrence [9, 11–14]. These clinical benefits might reflect results from several *in vitro* and translational experiments: 1. BCAA, in particular leucine, is able to stimulate hepatocyte growth factor production by hepatic stellate cells, which may facilitate liver regeneration [15], 2. BCAA modifies

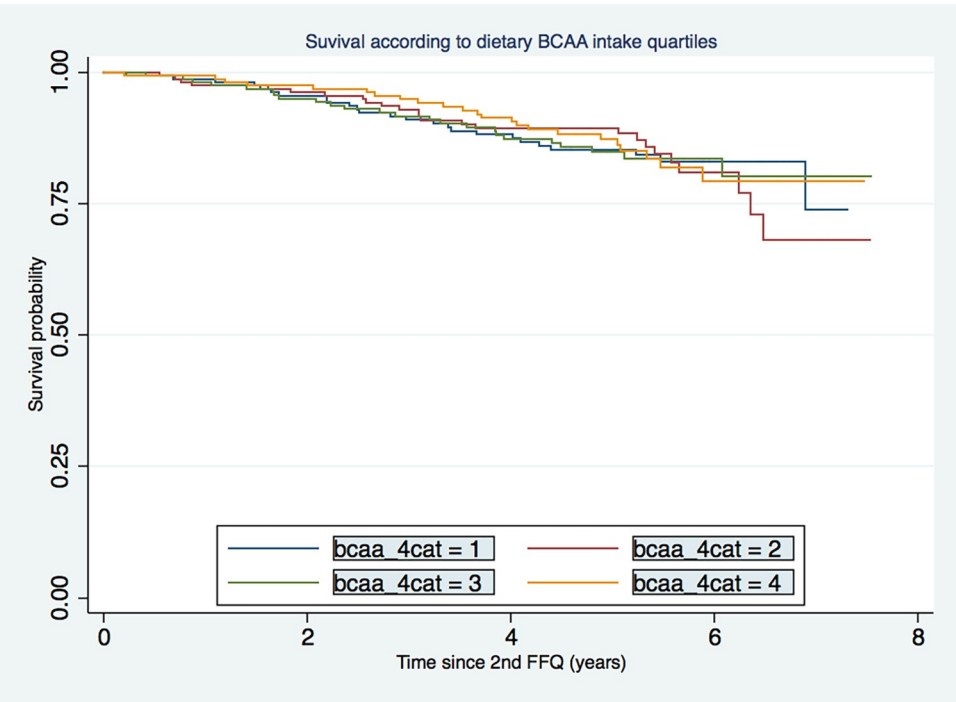

**Fig 1. Kaplan-Meier survival graph according to quartiles of dietary BCAA intake, censoring occurred when a patient either died from liver-related death or underwent liver transplantation.**

**Table 3. Risk of liver-related death or transplantation according to quartiles of average BCAA intake (measured in grams of BCAA per 1000 kcal of daily energy intake).**

| Quartiles and range (g/1000 kcal of energy) of BCAA intake | N | Person-years | Events (N) | | Incidence of liver-related death or transplant (per 1,000 person-years) | HR (95% CI) | [a]AHR (95% CI) | [a]P-value and AHR for trend |
|---|---|---|---|---|---|---|---|---|
| | | | Transplant | Liver- related death | | | | |
| 1 (<5.6) | 164 | 788 | 16 | 9 | 31.9 | 1.00 | 1.00 | |
| 2 (5.6–6.4) | 164 | 766 | 14 | 11 | 32.4 | 1.04 (0.60–1.81) | 0.92 (0.49–1.75) | |
| 3 (6.4–7.4) | 164 | 753 | 10 | 14 | 31.8 | 1.03 (0.59–1.80) | 0.97 (0.50–1.90) | |
| 4 (>7.4) | 164 | 767 | 12 | 11 | 29.9 | 0.97 (0.55–1.70) | 1.03 (0.52–2.05) | P = 0.89 1.02 (0.81–1.27) |

[a]Adjusted for age, sex, race, BMI, diabetes, lifetime alcohol intake, smoking status, coffee intake, self-reported health status, cirrhosis status, duration of infection, peginterferon treatment group, daily average energy intake and daily average cholesterol intake. Only subjects with complete data (N = 585) were included in the multiple Cox regression model.

the structure and improves the function of albumin, which may alleviate oxidative stress associated with advanced cirrhosis [31, 32] and 3. BCAA improves cirrhosis associated insulin resistance by inhibiting insulin mediated anti-apoptosis pathway in liver cancer cells [16, 33].

If BCAA does in fact have a benefit in patients with cirrhosis, two important differences between our study and others may explain our inability to detect it. The first is that the average BCAA exposure in the aforementioned clinical trials among those who received supplementation is substantially higher than even patients in the highest BCAA intake quartile (median intake 15.7 g/day) in our study. Assuming the median intake of BCAA from diet in these trials was similar to that of our study, at 11.9 g/day, supplementing 12 g/day of BCAA would increase one's total BCAA intake to 24 g/day, or by 100%. In contrast, the difference in median absolute BCAA intake between the highest (15.7 g/day) and the lowest (9.6 g/day) quartile in our study was 6.1 g/day, which was only half of the difference between those who were and were not supplemented in the clinical trials. The lack of benefit observed in our study may be related to the low absolute levels of BCAA intake and the small variance in intake across the quartiles (comparison groups). The second important difference is that all other studies included patients with cirrhosis at varying stages whereas nearly two thirds of patients in our study had histologically confirmed advanced fibrosis without cirrhosis. The median serum albumin across the quartiles of dietary BCAA was 3.9 g/dL, reflecting excellent liver function. Our study may not have sufficient power to detect differences in liver related death, transplantation or HCC which were less common in noncirrhotic patients [24, 34].

As in all observational studies, it is impossible to exclude unmeasured confounding despite our extensive multivariate adjustment. However, several strengths of our study are worth highlighting. First, estimation of BCAA intake was based on a validated FFQ administered twice prior to the onset of clinical events–therefore allowing more accurate capture of long-term intake of BCAA. Multiple asssessments have been shown to reduce measurement errors from recall bias associated with FFQ [26]. Second, we found no associations between liver related outcomes and three different models of dietary BCAA intake which, as we outlined in the Methods, have different biological interpretations. These negative sensitivity analyses suggest that there is unlikely a "missing" association at least in our study population. Finally, the current analysis represents the largest study (N = 656) with the longest follow-up (5.0 years)

on the relationship between BCAA and liver disease outcomes, and to our knowledge, the only study that assessed BCAA exposure from diet.

In conclusion, we did not find an association between the level of dietary BCAA intake and liver-related death or transplantation, development of HCC, hepatic encephalopathy or clinical decompensation in a well-characterized cohort of HCV-infected patients. Our result suggests that BCAA at relatively low doses is unlikely to influence liver disease progression in patients with early stage cirrhosis. Whether BCAA supplementation at higher doses can reduce the risk of liver cancer in well-compensated cirrhosis warrants further study.

## Trial information

Study data were derived from the extended follow-up data from the Hepatitis C Antiviral Long-term Treatment against Cirrhosis (HALT-C) Trial (ClinicalTrials.gov #NCT00006164).

## Supporting information

**S1 Table. Rates of liver related decompensations according to quartiles of BCAA intake derived from average daily BCAA intake (measured in grams of BCAA per 1000 kcal of daily energy intake).**
(DOCX)

**S2 Table. Crude and adjusted hazard ratios of liver-related death or transplantation according to quartiles of BCAA intake derived from average daily energy adjusted BCAA intake (measured in grams of BCAA per 1000 kcal of daily energy intake).**
(DOCX)

**S3 Table. Crude and adjusted hazard ratios of first liver related decompensations (including first event of variceal bleeding, ascites, spontaneous peritonitis or encephalopathy) according to quartiles of BCAA intake derived from average daily BCAA intake (measured in grams of BCAA per 1000 kcal of daily energy intake).**
(DOCX)

**S4 Table. Risk of liver-related death or transplantation according to quartiles of average BCAA/total protein/caloric intake (measured in grams BCAA per grams total protein per 1000 kcal).**
(DOCX)

**S5 Table. Risk of liver-related death or transplantation according to quartiles of average daily absolute BCAA intake without accounting for total energy intake (measured in grams of BCAA).**
(DOCX)

## Acknowledgments

The Hepatitis C Antiviral Long-term Treatment against Cirrhosis (HALT-C) Trial was conducted by the HALT-C Investigators and supported by the National Institute of Diabetes and Digestive and Kidney Diseases (NIDDK). The data from the HALT-C Trial reported here was supplied by the NIDDK Central Repository. This manuscript does not necessarily reflect the opinions or views of the HALT-C study, the NIDDK Central Repository or the NIDDK.

## Author Contributions

**Conceptualization:** Lei Yu, Shirley C. Paski, Jennifer Dodge, Kiran Bambha, Scott W. Biggins, George N. Ioannou.

**Investigation:** Lei Yu.

**Project administration:** Lei Yu.

**Resources:** George N. Ioannou.

**Supervision:** Lei Yu.

**Visualization:** Lei Yu.

**Writing – original draft:** Lei Yu.

**Writing – review & editing:** Shirley C. Paski, Jennifer Dodge, Kiran Bambha, Scott W. Biggins, George N. Ioannou.

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
