## [Decision Letter · Decision Letter 0]

26 Jul 2022

PONE-D-22-17427Effect of dietary branched chain amino acids on liver related mortality: Results from a large cohort of North American patients with advanced HCV infectionPLOS ONE

Dear Dr. Yu,

Thank you for submitting your manuscript to PLOS ONE. After careful consideration, we feel that it has merit but does not fully meet PLOS ONE’s publication criteria as it currently stands. Therefore, we invite you to submit a revised version of the manuscript that addresses the points raised during the review process.

We look forward to receiving your revised manuscript.

Kind regards,

Wan-Long Chuang, M.D., Ph.D.

Academic Editor

PLOS ONE

Journal Requirements:

3. PLOS requires an ORCID iD for the corresponding author in Editorial Manager on papers submitted after December 6th, 2016. Please ensure that you have an ORCID iD and that it is validated in Editorial Manager. To do this, go to ‘Update my Information’ (in the upper left-hand corner of the main menu), and click on the Fetch/Validate link next to the ORCID field. This will take you to the ORCID site and allow you to create a new iD or authenticate a pre-existing iD in Editorial Manager. Please see the following video for instructions on linking an ORCID iD to your Editorial Manager account: https://www.youtube.com/watch?v=_xcclfuvtxQ.

5. Please include your tables as part of your main manuscript and remove the individual files. Please note that supplementary tables should remain as separate "supporting information" files.

Reviewers' comments:

Reviewer's Responses to Questions

**Comments to the Author**

1. Is the manuscript technically sound, and do the data support the conclusions?

Reviewer #1: Partly

Reviewer #2: Partly

2. Has the statistical analysis been performed appropriately and rigorously? 

Reviewer #1: No

Reviewer #2: I Don't Know

3. Have the authors made all data underlying the findings in their manuscript fully available?

Reviewer #1: No

Reviewer #2: Yes

4. Is the manuscript presented in an intelligible fashion and written in standard English?

Reviewer #1: Yes

Reviewer #2: Yes

5. Review Comments to the Author

Reviewer #1: In this retrospective cohort study conducted by using the extended follow-up data from the HALT-C trial, the authors suggested that dietary BCAA intake was not associated with liver-related death or liver transplantation. Besides, incidence of HCC or hepatic encephalopathy was not associated with dietary BCAA level.

Major comments

The authors used the extended data from the HALT-C trial, and the patients enrolled in this trial were with hepatic fibrosis (Ishak score 3-4) or compensated cirrhosis (CTP score < 7). According to the previous studies, the effects on BCAAs, which act as pharmaco-nutrients but not medication, on hepatic outcomes were more suggested in patients who had developed cirrhotic complications. The less severity of liver disease in these patients may contribute to the negative findings in this study.

1. The dietary of BCAA was assessed by the food frequency questionnaire, the cornerstone of this study. The authors have to explain how to deal with the potential response bias and recall bias that would have the major influence in this study.

2. As mentioned in your method, the diet of patients may change overtime, and multiple assessments were applied to calculate the dietary components. However, the detailed information, such as the interval of questionnaire assessment, was not described at all. Considering the long-term follow-up period in your study, the authors have to provide more information about the questionnaire assessment and describe how to ensure the stability of the patients’ dietary habit/ component during the assessment interval. The authors also have to tell how to manage the data if a remarkable change of dietary habit/component was observed in a patient during the long-term follow-up.

3. The BCAA intake is positively correlated with total energy intake, which is closely associated with the whole nutritional status and would have substantial impacts on the clinical outcomes. Besides, the BCAA intake amount was also found significantly associated with the dietary amount of aromatic AA that would also affect the presence of HE and other complications. To reduce the overfitting and multicollinearity of these factors, a regularized regression analysis, such as Ridge regression or Lass regression, is more appropriate to investigate the risk of outcomes in this study.

Minor comments

1. The primary and secondary outcomes of this study included liver-related death, liver transplantation, presence of HCC and HE. Considering the enrolled patients were only with advanced fibrosis or compensated cirrhosis, the incidence of hepatic decompensation, including ascites formation, spontaneous bacterial peritonitis, variceal bleeding, and so on is suggested to investigate. Besides, the definition of liver-related death is needed to be clarified.

2. The crude hazard ratio of each factor that had been input into the regression model for the analyses of the risk of liver-related death, liver transplantation or liver decompensation should be provided in the table. The crude and adjusted risks of each factor to the presence of hepatic decompensation or HCC are also needed to be provided in the separate tables.

Reviewer #2: General comments

The study by Yu, et al. evaluated retrospectively the effect of dietary branch chain amino acids (BACC) on the liver- related outcomes including liver-related death, liver transplantation, hepatoma (HCC) development and hepatic encephalopathy (HE) in a HALT-C cohort of patients with advanced fibrosis or cirrhosis. In a cohort of 656 patients stratified into four groups by BCAA intake, there are no significant differences of the incidences of liver-related death, liver transplantation, HCC development and HE in a median follow-up of 5 years. Contrary to previous studies using BCAA as supplements of diet, this study estimated the BCAA contents of patients’ regular diet to investigate its impact on liver-related outcomes. Although interesting, there are some issues to be clarified.

Major comments

1. It is important to estimate of BCAA amount from the patients’ diet in this study. In the method section, the author might explain briefly how they calculated the BCAA amount from patients’ diet in the food questionnaires.

2. Are the incidences of liver- related outcomes comparable to the similar patient cohorts outside HALT-C trial? This issue might be provided in the discussion section.

3. In addition to liver-related outcomes, the readers might also be interested in the all-cause and non-liver related death. The authors might provide these data.

4. Direct-acting antivirals (DAA) will have a great impact on the liver-related outcomes. Are these patients undergoing DAA for hepatitis C virus treatment?

5. Whether sarcopenia has an impact of the liver-related outcomes in this cohort might be analyzed in this study.

6. PLOS authors have the option to publish the peer review history of their article (what does this mean?). If published, this will include your full peer review and any attached files.

Reviewer #1: No

Reviewer #2: No

---

## [Author Response · Author response to Decision Letter 0]

9 Feb 2023

response letter attached as new cover letter.

---

## [Decision Letter · Decision Letter 1]

27 Feb 2023

PONE-D-22-17427R1Effect of dietary branched chain amino acids on liver related mortality: Results from a large cohort of North American patients with advanced HCV infectionPLOS ONE

Dear Dr. Yu,

Thank you for submitting your manuscript to PLOS ONE. After careful consideration, we feel that it has merit but does not fully meet PLOS ONE’s publication criteria as it currently stands. Therefore, we invite you to submit a revised version of the manuscript that addresses the points raised during the review process.

We look forward to receiving your revised manuscript.

Kind regards,

Wan-Long Chuang, M.D., Ph.D.

Academic Editor

PLOS ONE

Journal Requirements:

Reviewers' comments:

Reviewer's Responses to Questions

**Comments to the Author**

1. If the authors have adequately addressed your comments raised in a previous round of review and you feel that this manuscript is now acceptable for publication, you may indicate that here to bypass the “Comments to the Author” section, enter your conflict of interest statement in the “Confidential to Editor” section, and submit your "Accept" recommendation.

Reviewer #1: All comments have been addressed

Reviewer #2: All comments have been addressed

Reviewer #3: (No Response)

2. Is the manuscript technically sound, and do the data support the conclusions?

Reviewer #1: Yes

Reviewer #2: (No Response)

Reviewer #3: Yes

3. Has the statistical analysis been performed appropriately and rigorously? 

Reviewer #1: Yes

Reviewer #2: (No Response)

Reviewer #3: Yes

4. Have the authors made all data underlying the findings in their manuscript fully available?

Reviewer #1: Yes

Reviewer #2: (No Response)

Reviewer #3: Yes

5. Is the manuscript presented in an intelligible fashion and written in standard English?

Reviewer #1: Yes

Reviewer #2: (No Response)

Reviewer #3: Yes

6. Review Comments to the Author

Reviewer #1: In the current manuscript, the authors had answered my question and revised the text appropriately.

It is ready to be accepted.

Reviewer #2: (No Response)

Reviewer #3: PONE-D-22-17427R1: statistical reviiew

SUMMARY. This is a retrospective cohort study of the association between dietary BCAA intake and the time up liver-related death or liver transplantation. The effect of BCAA levels on some secondary ouctomes is also investigated. The core statistical analysis correctly relies on a battery of Cox regression models where the effect of interest is estimated, by adjusting for the available confounders. The methods are correct and the limitations of the study are well described in the discussion. I just list below two minor points that the authors should be able to address without too much effort.

MINOR ISSUES

1. Line 159: "There was no significant change in median BCAA intake estimated from the follow-up FFQ (6.4 g/1000 kcal, IQR 5.4 – 7.5) compared to that from baseline (6.4 g, IQR 5.4 – 7.3 g)." I'd welcome a summary of the distribution of the individual differences between baseline and follow-up intakes. This would indicate that dietary behavior is essentially time-constant, motivating its use as a baseline covariate in the Cox model.

2. Line 128: ". Normality for continuous variables were assessed using the Shapiro-Wilk-Test." This sentence can be removed: covariates do not need to be normally distributed.

7. PLOS authors have the option to publish the peer review history of their article (what does this mean?). If published, this will include your full peer review and any attached files.

Reviewer #1: No

Reviewer #2: No

Reviewer #3: No

---

## [Decision Letter · Decision Letter 2]

10 Apr 2023

Effect of dietary branched chain amino acids on liver related mortality: Results from a large cohort of North American patients with advanced HCV infection

PONE-D-22-17427R2

Dear Dr. Yu,

We’re pleased to inform you that your manuscript has been judged scientifically suitable for publication and will be formally accepted for publication once it meets all outstanding technical requirements.

Kind regards,

Wan-Long Chuang, M.D., Ph.D.

Academic Editor

PLOS ONE

Additional Editor Comments (optional):

Reviewers' comments:

Reviewer's Responses to Questions

**Comments to the Author**

1. If the authors have adequately addressed your comments raised in a previous round of review and you feel that this manuscript is now acceptable for publication, you may indicate that here to bypass the “Comments to the Author” section, enter your conflict of interest statement in the “Confidential to Editor” section, and submit your "Accept" recommendation.

Reviewer #1: All comments have been addressed

Reviewer #2: (No Response)

Reviewer #3: All comments have been addressed

2. Is the manuscript technically sound, and do the data support the conclusions?

Reviewer #1: Yes

Reviewer #2: (No Response)

Reviewer #3: (No Response)

3. Has the statistical analysis been performed appropriately and rigorously? 

Reviewer #1: Yes

Reviewer #2: (No Response)

Reviewer #3: (No Response)

4. Have the authors made all data underlying the findings in their manuscript fully available?

Reviewer #1: Yes

Reviewer #2: (No Response)

Reviewer #3: (No Response)

5. Is the manuscript presented in an intelligible fashion and written in standard English?

Reviewer #1: Yes

Reviewer #2: (No Response)

Reviewer #3: (No Response)

6. Review Comments to the Author

Reviewer #1: The authors had revised the manuscript well. All the comments I raised had been addressed accordingly. I think it is ready to be accepted.

Reviewer #2: (No Response)

Reviewer #3: (No Response)

7. PLOS authors have the option to publish the peer review history of their article (what does this mean?). If published, this will include your full peer review and any attached files.

Reviewer #1: No

Reviewer #2: No

Reviewer #3: No

---

## [Editor Report · Acceptance letter]

13 Apr 2023

PONE-D-22-17427R2 

Effect of dietary branched chain amino acids on liver related mortality: Results from a large cohort of North American patients with advanced HCV infection 

Dear Dr. Yu:

I'm pleased to inform you that your manuscript has been deemed suitable for publication in PLOS ONE. Congratulations! Your manuscript is now with our production department. 

Kind regards, 

on behalf of

Dr. Wan-Long Chuang 

Academic Editor

PLOS ONE